# Widespread Resistance to Temephos in *Aedes aegypti* (Diptera: Culicidae) from Mexico

**DOI:** 10.3390/insects15020120

**Published:** 2024-02-07

**Authors:** Jesus A. Davila-Barboza, Selene M. Gutierrez-Rodriguez, Alan E. Juache-Villagrana, Beatriz Lopez-Monroy, Adriana E. Flores

**Affiliations:** Facultad de Ciencias Biologicas, Universidad Autonoma de Nuevo Leon, Av. Universidad s/n Cd. Universitaria, San Nicolas de los Garza 66455, Mexico; jdavilab@uanl.edu.mx (J.A.D.-B.); selene.gutierrezro@uanl.edu.mx (S.M.G.-R.); alan.juache@gmail.com (A.E.J.-V.); beatriz.lopezmr@uanl.edu.mx (B.L.-M.)

**Keywords:** larvicide, temephos, *Aedes aegypti*, intensity of resistance

## Abstract

**Simple Summary:**

In Mexico, the primary method for managing arboviruses transmitted by *Aedes aegypti* involves the utilization of insecticides targeting both larval and adult stages. Temephos, a larvicide, has been a cornerstone in vector control strategies for over five decades. Our comprehensive analysis of twenty-three *Ae. aegypti* populations across Mexico revealed a widespread and moderately intense resistance to temephos. This emerging resistance is a cause for concern, signaling a potential decline in the efficacy of temephos, a crucial component in mosquito population management and disease prevention. Our study underscores the urgent need to explore and implement alternative mosquito control strategies to maintain effective public health interventions. These findings strongly indicate that the future of mosquito control, particularly in larval stages, may no longer rely on temephos alone, thereby emphasizing the imperative for the innovation and adoption of novel larvicidal methods or agents.

**Abstract:**

Organic synthetic insecticides continue to be part of the arsenal for combating vector-borne diseases in Mexico. Larvicides are a fundamental part of the process in programs for mosquito control, temephos being one of the most widely used in Mexico. In the present study, we analyzed the frequency of temephos resistance in twenty-three *Aedes aegypti* populations using the discriminating concentration (DC) of 0.012 mg/L. We also tested 5× DC (0.6 mg/L) and 10× DC (0.12 mg/L) of temephos. The resistance distribution to temephos was interpolated to unsampled sites using the inverse distance weighting (IDW) method. The populations of *Ae. aegypti* showed a high frequency of resistance (1× DC) with mortality rates below 93% in 22 of the 23 populations analyzed. Moderate resistance intensity (5× DC) was found in 78% of the populations, and high intensity (10× DC) in 30%. Predicted mortality was below 60% in the populations of the Pacific Coast, along the Gulf of Mexico, and in the state of Coahuila in Northeastern Mexico in relation to 1× DC; the Pacific Coast and Northeast patterns hold for 5× and 10× DC. The results suggest the need for rotation of the larvicide to effectively control the larval populations of the vector in the country.

## 1. Introduction

Dengue, chikungunya, and Zika arboviruses persist as a threat to public health in many parts of the world, including Mexico. The transmission of these pathogens is mediated through the infectious bites of several mosquito species, emphasizing the role of *Ae. aegypti* as the primary vector [1]. In Mexico, based on current regulations on epidemiological surveillance, promotion, prevention, and control of vector-borne diseases (NOM-032-SSA2-2014), the CENAPRECE (National Center of Preventive Programs and Disease Control) recommends several strategies to diminish the risk of arboviral transmission by reducing the populations of vector mosquitoes such as *Ae. aegypti* and issues the list of recommended supplies (insecticides and application equipment) for this purpose. These insecticides approved for public health must meet a series of requirements, highlighting the susceptibility to these in the vector populations to be controlled. Although the CENAPRECE includes larvicides such as *Bacillus thuringiensis* var *israelensis* (*Bti*), methoprene, spinosad, and novaluron in the list of insecticides approved for use in Mexico, temephos is a larvicide that appears year after year [2,3].

The use of temephos has been recorded since 1969 in Australia, Canada, Ecuador, the United States, and Mexico, among other countries [4]. Resistance to this insecticide is found in several populations of *Ae. aegypti* from the Americas. It has been reported in Brazil by Chediak et al. [5], who observed that larval mortality to temephos has decreased below 80% since 2004. Resistance has also been documented in Colombia, where the rates of mortality were less than 80% just after two weeks of the application of a field dose of temephos, and it dropped to less than 50% after four weeks; they also estimated that the LC_50_ of a field population was 15× higher compared to susceptible lines [6]. This same pattern was also found in Peru, where 27 out of 39 populations of *Ae. aegypti* displayed moderate to high temephos resistance [7]. Few studies have been conducted to assess the susceptibility of *Ae. aegypti* to the larvicide temephos in Mexico. The first study revealed that larvae collected from three locations in the state of Guerrero in Southwest Mexico were susceptible to temephos, evaluated with dose–response bioassays [8]. Another study revealed that larvae collected from two distinct locations in Chiapas, in the southern region of Mexico, exhibited resistance, with mortality rates ranging from 42% to 78% when exposed to the DC of 0.012 mg/L of temephos [9]. On the other hand, Ciau-Mendoza et al. [10] identified a low-intensity resistance to temephos in larvae sourced from four different locations in Quintana Roo, Southeast Mexico.

This study aimed to determine the frequency of resistance to temephos in 23 populations of *Ae. aegypti* from Mexico through bioassays using the discriminating concentration of 0.012 mg/L proposed by WHO [11]. We also determined the intensity of resistance by increasing this established concentration 5 and 10 times. The spatial distribution of temephos resistance was interpolated to unsampled sites using the inverse distance weighting (IDW) method.

## 2. Materials and Methods

### 2.1. Collection Sites and Maintenance of Colonies

The populations of *Ae. aegypti* were obtained during 2019–2020 directly from the field in larval stages. From Northeast Mexico in the state of Nuevo Leon in two locations, Monterrey and Guadalupe, and the state of Coahuila in two locations, Piedras Negras and Muzquiz. In the Central North of the country, the sites were in the states of San Luis Potosi in the city of San Luis Potosi, the state of Aguascalientes in the city of Aguascalientes, and the state of Guanajuato in the city of Leon. On the Gulf of Mexico coast, sampling locations were in the state of Veracruz in Coatzacoalcos, Cosoleacaque, Panuco, Tuxpan, and Acayucan. Other sites were in the Southeastern region in the state of Yucatan, in Temax, Valladolid, and Izamal, and the Southwest, in the state of Guerrero in the city of Zihuatanejo. In western Mexico, in Jose Maria Morelos in the municipality of Tomalan, Lagos de Moreno, and Puerto Vallarta in the state of Jalisco. Finally, in the Northwest, other sites were in the state of Sinaloa in the city of Mazatlan and the state of Baja California Sur in the cities of La Paz and Loreto (Figure 1).

Directed samplings were carried out, collecting the larval stages in at least ten developmental sites per location (clean stored or stagnant water), including drums, vases, pools, and tires. The larvae were collected using standard probes. The material was then placed in Whirl-Pack bags (BioQuip Products, Inc. Glandwick, CA, USA), duly labeled, and transported to the insectary of the Insect Physiology and Toxicology area of the Medical Entomology Laboratory of the Facultad de Ciencias Biologicas, UANL, Mexico.

At each collection site, between 1200 and 1500 larvae were obtained, and they were placed in plastic trays identified with the name of each sampling location containing dechlorinated water. Larvae were fed with powdered bovine liver protein (Liver Powder MP Biomedicals, LLC, Santa Ana, CA, USA). Once all collected larvae reached the pupal stage, they were transferred to emergence chambers and placed in 30 × 30 cm cages. The biological material obtained from the field was reared under insectary conditions at 28 ± 1 °C and 70–80% relative humidity with a 12 h:12 h (light:dark) photoperiod.

We obtained at least 600 adults from each site, which were maintained in sets of 200 adults per cage. Male mosquitoes were fed with 10% sugar water on impregnated cotton, and females by artificial feeding with lamb’s blood (*Ovis orientalis*) for egg production. Inside the cages, plastic cups with dechlorinated water and filter paper were placed as a substrate for oviposition. Once the females (F_0_) laid the eggs (F_1_), they were reserved for subsequent bioassays. The papers with the eggs (F_1_) were placed in trays with dechlorinated water, and a pinch of powder yeast (*Saccharomyces cerevisiae* yeast, Sigma Aldrich, St. Louis, MO, USA) was added to stimulate the hatching of the eggs. This was kept within a controlled light incubator maintained at 28 ± 1 °C (Prendo INLC-20, Puebla, Mexico).

### 2.2. Bioassays

Bioassays were performed on late 3rd instar/early 4th instar larvae (F_1_). The active ingredient temephos (technical grade, 97.5% purity; Chem Service, West Chester, PA, USA) was used, which was diluted in ethanol to obtain a 100 mg/mL stock, from which the discriminating concentration (DC) of 0.012 mg/L was obtained when diluted in distilled water to a volume of 100 mL [11].

Late 3rd/early 4th instar larvae were exposed to DC of 0.012 mg/L in groups of 25 individuals per replicate (4 replicates) with a control containing 1 mL of ethanol diluted in water (25 individuals). Mortality was recorded at 24 h, and Abbott’s formula was applied when mortality was found in the control between 5 and 20%, and the bioassay was discarded when control mortality was over 20% [12]. All procedures described above were also performed on the New Orleans (NO) reference susceptible strain.

The frequency of resistance was calculated using the WHO criteria to categorize populations as follows: susceptible when mortality was ≥98%; between 90 and 97% suggests the possibility of resistance, requiring confirmation; and mortality < 90% indicates resistance [13]. If the bioassay with DC of 0.012 mg/L detected resistance, bioassays were carried out with DC increased five times 0.06 mg/L (5× DC) and ten times 0.12 mg/L (10× DC). Resistance intensity results were interpreted as follows: ≥98% mortality at 5× DC exposure was considered low intensity, and <98% mortality was moderate to high intensity. For the case of exposure to 10× the DD, mortality ≥ 98% was considered moderate intensity resistance, and <98% mortality was high-intensity resistance [13]. On all occasions when the mortality percentages fell within the decision limits in a bioassay, that is, 90% or 98%, the bioassay was repeated.

The mortality percentages recorded for each population and discriminating concentrations (1× DC, 5× DC, and 10× DC) were compared between populations using Kruskal–Wallis multiple comparisons (α < 0.05) (GraphPad Version 8, GraphPad Software, La Jolla, CA, USA).

### 2.3. Distribution of Resistance to Temephos

The temephos resistance distribution was analyzed using the inverse distance weighting (IDW) interpolation procedure. The mortality in larvae when exposed at 1× DC, 5× DC, and 10× DC, and the distance between sampled sites were used to estimate the mortality for each temephos concentration in the unsampled regions. The interpolation was generated in a buffer of 200 km from each sampling site to allow for the gene flow of *Ae. aegypti*, as reported by Gorrochotegui et al. [14,15].

The IDW method assumes that the distance between elements determines the values of unmeasured points; the closer points will have more similar values than the distant ones [16]. The power parameter of the IDW formula controls the degree of influence of closer points in the interpolated values. Given this, it is necessary to identify the power value that results in the best estimates. Four different values (2, 2.5, 3, 3.5, and 4) were used; known sampled locations were then interpolated, and the prediction estimates were compared to the measured values to obtain the Root Mean Square Error (RMSE) as a measure of the fit. The power value with the lower RMSE was selected as it produces the best prediction estimates. All analyses were performed in R version 4.1.0 and QGIS.

## 3. Results

### 3.1. Frequency and Intensity of Resistance to Temephos

Susceptibility to temephos was analyzed in 23 populations of *Ae. aegypti*, of which 96% were resistant when exposed to 1× DC of temephos (22/23), with mortalities between 3 and 93%; only the Izamal, Yucatan population was susceptible, with 100% mortality. All populations were subjected to 5× DC of temephos, and 78% (18/23) showed a moderate or high intensity of resistance (mortality < 98%). The populations that showed the low intensity of resistance with mortalities between 98 and 100% were San Luis Potosi from the state of San Luis Potosi, Cosoleacaque from Veracruz; Izamal from Yucatan; Lagos de Moreno from Jalisco, and La Paz from Baja California Sur.

Moderate intensity of resistance (mortality ≥ 98%) was confirmed in 70% of the populations (16/23). In contrast, the populations of Muzquiz from Coahuila, Coatzacoalcos, and Acayucan from Veracruz, Temax from Yucatan, Zihuatanejo from Guerrero, Jose Maria Morelos from Jalisco, and Loreto from Baja California Sur, all showed a high intensity of resistance (mortality < 98%) according to the WHO criteria (2016) (Figure 2).

Additionally, the mortality percentages were compared between each population and for each discriminating concentration (1× DC, 5× DC, and 10× DC), and the populations that showed both significantly higher frequency and intensity of resistance were Zihuatanejo, from Guerrero, on the Pacific Coast; Muzquiz from Coahuila, in the northeast; and Acayucan from Veracruz, on the east coast (*p* < 0.05) (Figure 2).

### 3.2. Distribution of Resistance to Temephos

For 1× DC, the power value of three resulted in the lowest RMSE value (0.7426). Figure 3A shows the trends indicating a significant part of the Gulf of Mexico (Veracruz), the coasts, and the Pacific coast states (Baja California Sur, Sinaloa, Jalisco, and Guerrero) with predicted mortalities below 60%. This pattern is also present in the state of Coahuila. The central region of Mexico and Yucatan display greater mortalities but remain below 98%.

For 5× DC, the power value of two resulted in the lowest RMSE value (0.4724). These results show that in central Mexico, mortalities were estimated to be 90 to 100%. However, the same resistance pattern was displayed for most of the Pacific and Gulf of Mexico coasts (Figure 3B). Finally, mortality with 10× DC of temephos is almost fully recovered in all predicted areas, as shown by the power value of 2.5, with the lowest RMSE value (0.6779). Nonetheless, two areas remained with high intensity of temephos resistance, corresponding to the vicinity of the states of Coahuila and Guerrero (Figure 3C).

## 4. Discussion

This study is the first approach to mapping insecticide resistance to temephos in Mexican populations of *Ae. aegypti*. The primary trend obtained from IDW interpolation was the increase in mortality with higher temephos concentrations. However, mortality does not reach the 98% cut-off for some zones. It is important to note that the distribution results include a biologically relevant parameter that allows for the distance at which gene flow occurs between populations [14,15]. However, the results of the IDW interpolation should be analyzed carefully because of the limitations of the technique. Even though it is straightforward and widely implemented, IDW interpolation possesses several limitations. For example, this procedure is deterministic, and variances of predicted, unmeasured sites are not calculated. Another technical drawback is its reliance on the parameter that controls the influence of proximate points (power value). It remains constant to the whole spatial extent and cannot dynamically adjust to varying sites [17]. Also, the distribution of insecticide resistance depends on other variables, such as insecticide application.

Although this work represents the first extensive study on resistance to temephos in Mexico, which included 23 populations of *Ae. aegypti* from all regions of the country, there were already two reports of resistance [9,10]. However, the results of one of the studies show a low intensity of resistance to temephos in four populations of *Ae. aegypti* from Quintana Roo, Southeastern Mexico, contrary to our results in which less than 78% (18/23) of the populations showed a moderate intensity of resistance, and of these populations, 39% (7/18) had a high resistance intensity [10]. A larger-scale study shows a clearer picture of the temephos resistance status of *Ae. aegypti* in the country, which could guide decision-making for using this larvicide in control programs for this vector. In this regard, studies in Mexico have demonstrated the low efficacy of temephos in the field compared to other larvicides, like the study by Marina et al. [18], who evaluated the efficacy of spinosad against mosquito larvae in used car tires, comparing its performance to temephos granules and a *Bti*-based product. They demonstrated that spinosad treatments displayed superior and longer-lasting control of *Ae. aegypti*, *Ae. albopictus*, *Culex quinquefasiatus*, and *Cx. coronator* larvae, with effectiveness lasting 6–8 weeks; meanwhile, temephos provided an intermediate duration of control, showing effectiveness for approximately three weeks before mosquito populations began to rebound. In this study, *Bti* performance was relatively poor. This suggests that while temephos was effective, alternatives like spinosad may offer more sustained mosquito population control in specific environments such as car tires.

In more recent studies, the effectiveness of various larvicides against *Ae. aegypti* in insecticide-treated ovitraps was evaluated in the city of Chiapas in southern Mexico amid outbreaks of chikungunya. They found that insecticide-treated ovitraps with spinosad and λ-cyhalothrin significantly reduced *Ae. aegypti* larvae and pupae for more extended periods than temephos-treated ovitraps [19]. Coinciding with another study developed in used vehicle tires in Veracruz, Mexico, which highlights the high effectiveness of λ-cyhalothrin and spinosad in the control of *Aedes* spp over an extended period of 9–12 weeks and the null performance of temephos, suggesting a reduced susceptibility to the latter in *Aedes* populations [20]. Other studies have also demonstrated the low effectiveness of temephos, which only provided control of *Ae. aegypti* larvae in ovitraps and domestic water tanks in Chiapas in southern Mexico for up to 3 weeks compared to spinosad and novaluron, which provided 7 to 12 weeks of control [21].

Temephos is an efficient and low-cost control agent, which has led to its widespread use as a larvicide in many areas of the world, thereby resulting in the development of resistance in various mosquito populations [22,23]. This organophosphate was used for 30 years prior to the first report of resistance in 1995 in *Ae. aegypti* in Venezuela, and despite various records of resistance, temephos remains one of the most widely used larvicides in tropical regions worldwide [24].

From the record of resistance to temephos in Venezuela and several Caribbean countries, the determination of the resistance status to improve vector control has increased [24,25]. Since 2000, resistance to temephos has been reported in Cuba and Venezuela, Panama, Brazil, El Salvador, the island of Martinique in the French Antilles, Argentina, Colombia, Trinidad, Costa Rica, and Ecuador [5,6,26,27,28,29,30,31,32,33,34,35,36,37,38,39,40,41,42,43,44]. More recent reports have come from Peru, Mexico, and other non-Latin American countries such as Thailand, Malaysia, and India [7,9,10,45,46,47,48,49]. However, some of these studies report the susceptibility of *Ae. aegypti* to temephos, despite the prolonged use of this larvicide, such as in Venezuela, where Alvarez et al. [50] determined that populations from western Venezuela were susceptible to temephos with resistance ratios of RR < 5. Similar results were reported for *Ae. aegypti* from Puerto Rico [51].

Even though this larvicide has been used for more than 50 years in that country, a recent study in the Colombian Caribbean region showed susceptibility in *Ae. aegypti* from fifteen municipalities, with 98% and 100% mortalities in bioassays with DC of 0.012 mg/L and resistance ratios (RR_50_ and RR_95_) of <5 in dose–response bioassays [52]. The authors considered that the susceptibility of the populations was due to the alternating use of temephos with growth regulators for the control of *Ae. aegypti*, which had a positive impact, maintaining temephos susceptibility in the populations of that region. Other studies report tolerance to temephos in populations of *Ae. aegypti* in Indonesia, with 92% mortality when exposed to DD [53]. In our study, only one population, Lagos de Moreno, Jalisco, exhibited mortality of 93% at 1× DC; however, when evaluated at 5× DC, the mortality of this population was 100%, thus confirming a low intensity of resistance according to the WHO criteria [13].

In Mexico, temephos has been used in dengue control campaigns for more than half a century [4] and continuously for mosquito control. Therefore, it is relevant to evaluate the impact of the prolonged use of this larvicide in the control campaigns, as assessed by Valle et al. [35], who stated that the prolonged use of temephos since 1967 in Brazil caused a progressive increase in resistance in *Ae. aegypti* from 1998 to 2017.

It is important to consider that we established susceptibility based on the use of the DC of 0.012 mg/L reported by the WHO [11], so establishing susceptibility through dose–response bioassays would give us a more extensive overview of the degree of susceptibility of the populations of *Ae. aegypti*. In our study, we only showed the results of resistance to temephos in populations of *Ae. aegypti* at a spatial level; they reveal the impact of prolonged use of this larvicide on vector control in the country. This emphasizes the importance of monitoring resistance in space and time, as indicated by Chediak et al. [5], who confirmed the impact of prolonged use of temephos in Brazil. They demonstrated the temporal spread and spatial scope of resistance to temephos, such as the decrease in average mortality from 80.31% between 1999 and 2000 to less than 50% between 2010 and 2011, observing a trend of increasing frequency of resistance, which coincides with failures in vector control in the field.

The variation in the frequency and intensity of resistance observed in the populations included in the study reflects the pressure exerted on the vector control campaigns in the country. A study demonstrated the high heritability of resistance to temephos in two populations of *Ae. aegypti* in the state of Quintana Roo when sequentially selected for five generations in the laboratory [22]. In India, Adhikari and Khanikor [23] studied a temephos-susceptible population of *Ae. aegypti* reared in the laboratory and continuously exposed fourth instar larvae to temephos for 28 generations. As a result of this exposure, they observed that compared to the F_0_ generation, resistance increased by 1.65 and 7.83 times from the F_4_ to F_28_ generation, respectively. Although we did not evaluate sequential exposure to temephos for generations, the results showed that 30% of the populations (Muzquiz, Coahuila; Coatzacoalcos and Acayucan, Veracruz; Temax, Yucatan; Zihuatanejo, Guerrero; Jose Maria Morelos, Jalisco and Loreto, Baja California Sur) had a high intensity of resistance, as there were individuals that survived up to 10 times the DC (10× DC).

In a more recent study, susceptibility to temephos has been reported in populations of *Aedes albopictus* in India, considering that this is a country in which larval control is subject to the use of this organophosphate [54]. Even though our study yielded results in which resistance predominates in most populations, we only recorded one population, Izamal, Yucatan (4.4%), whose mortality was 100% after exposure to 1× DC, being the only population that was fully susceptible to temephos.

In Mexico, the Entomological Research and Bioassay Units (UIEBs), which are part of the state health secretariats, play a critical role in ensuring the efficacy of insecticides against *Ae. aegypti*. These units are tasked with evaluating the biological effectiveness of commercial insecticides authorized for use by the CENAPRECE.

Mexican regulations, specifically NOM-032-SSA2-2014, mandate that synthetic larvicides must achieve acute mortality rates of over 98% within 24 h of application and demonstrate a residual effect of at least two months, maintaining 80% or higher mortality rates. In the case of insect growth regulators, acute mortality > 98% and/or emergence inhibition > 90% is considered acceptable, with a residual effect of more than three weeks, in which the effect of mortality or inhibition of emergence should be >80%. However, in the latest assessment of the biological effectiveness of larvicides conducted in 2021, with the contribution of the UIEBs of 29 states, some commercial formulations of temephos (in its granular form, which is the most widely used in the country) showed comparatively lower performance with respect to other larvicides such as *Bti* and spinosad, based on the averaged results of the participating UIEBs. In this assessment, the larvicides methoprene, novaluron, and pyriproxyfen induced inhibition of emergence greater than 90% up to 60 days [55].

Despite the process undertaken by the UIEBs, these assessments do not extend to analyzing the resistance of *Ae. aegypti* populations within each state. Following the evaluations of the biological effectiveness of commercial larvicides, CENAPRECE issues recommendations to the states on the most effective larvicides for use. Nevertheless, the final selection of larvicides by each state is heavily influenced by budget constraints, indicating a complex decision-making process that balances efficacy, cost, and public health needs.

Implementing larvicide rotation strategies as part of integrated vector management (IVM) programs offers a promising approach to mitigate resistance development in *Ae. aegypti* populations, crucial for controlling vector-borne diseases. Studies underscore the importance of rotating larvicides with distinct modes of action (MoA) to prevent or delay resistance, highlighting the need for continuous monitoring and local adaptation of these strategies [56].

Monitoring insecticide resistance in the field is critical for understanding the resistance status of field populations; however, determining resistance mechanisms, such as detoxifying enzymes or insensitivity in the target site, gives us a clear idea of the selection pressure exerted in mosquito populations. Thus, knowledge of the mechanisms through biochemical tests, the use of synergists, and molecular tests are necessary to establish the mechanisms associated with the high frequency and intensity of resistance to temephos in the populations of *Ae. aegypti* studied.

## 5. Conclusions

A high frequency of resistance to temephos is recorded in *Ae. aegypti* in Mexico, with a moderate to high intensity of resistance. These outcomes underscore the need for persistent surveillance of resistance patterns, the strategic rotation of larvicides, and the introduction and application of innovative approaches for managing the vector’s larval stages nationwide. 

## Figures and Tables

**Figure 1 insects-15-00120-f001:**
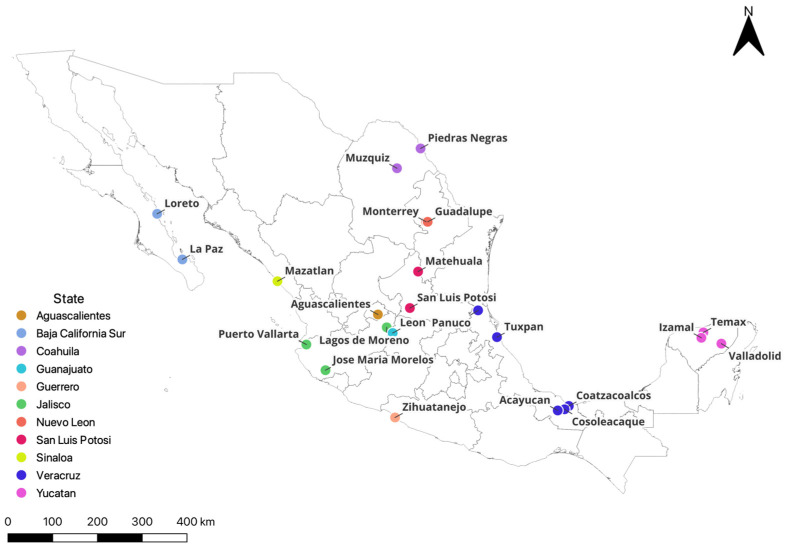
Sampling sites of larval populations of *Aedes aegypti* in Mexico. Different colors indicate the state. In some states, more than one site was sampled.

**Figure 2 insects-15-00120-f002:**
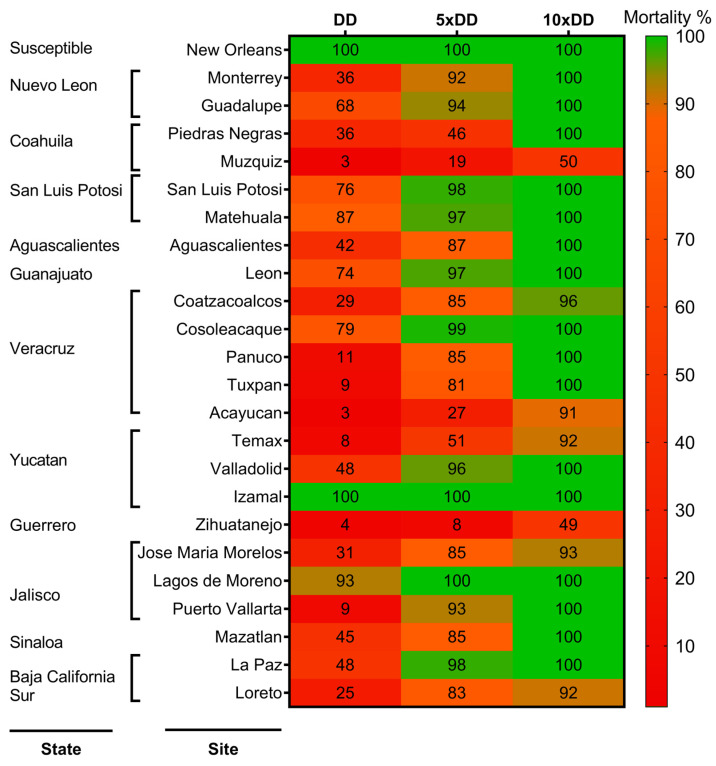
Heat map of frequency of resistance to 1× DC of temephos (0.012 mg/L) and intensity of resistance at 5× DC (0.06 mg/L) and 10× DC (0.12 mg/L) in larval populations of *Aedes aegypti* from Mexico. The percentage of mortality in each population is indicated at each dose. The susceptible reference strain (New Orleans) is also shown.

**Figure 3 insects-15-00120-f003:**
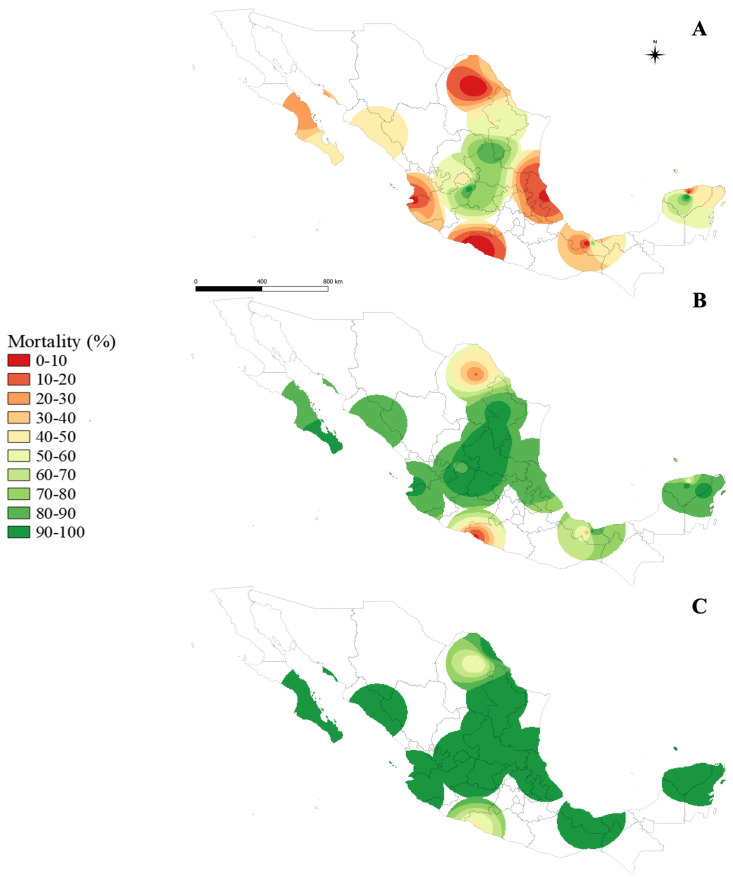
Distribution of temephos resistance (% of mortality to DC of 0.012 mg/L) in *Aedes aegypti* to 1× DC (0.012 mg/L) (**A**), 5× DD (0.06 mg/L) (**B**), and 10× DC (0.12 mg/L) (**C**) using IDW interpolation.

## Data Availability

All the required data relevant to the presented study are included in the manuscript.

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
