# Peer review of "Widespread Resistance to Temephos in Aedes aegypti (Diptera: Culicidae) from Mexico"

_insects, 2024, doi:10.3390/insects15020120_

Round 1
Reviewer 1 Report
Comments and Suggestions for Authors
Davila-Barbosa et al. report the results of a national-scale sampling study in which widescale resistance to temephos was detected in Mexico. This is a valuable study and will prove useful to public health workers in this region. I have numbered points and written suggestions on a scanned copy of the manuscript:
Numbered points (see scanned file):
1. Change to "...in insects exposed to the 1xDD;"
2. I did a Google search and found that reduced mortality of Aedes spp. exposed to temephos was previously reported in Veracruz in a comparative study on spinosad and lambda-cyhalothrin in used car tires published in Insects in 2019.
3. Could you provide an idea of how many adults were used to rear each batch of insects for bioassays. Obviously there could be an important genetic bottleneck here.
4. Did the control comprise diluted ethanol or just water?
5a. If >98% mortality was considered moderate resistance, this would include the 100% mortality result. I think you need to report this as a range e.g. 98-99%.
5b. Same issue, at 10xDD; >98% mortality would include a 100% mortality result, so you need to report a mortality range here.
6. Mortality rates? For clarity, you mean "percentage of mortality values" were subjected to statistical analysis, correct?
However, there were no Kruskal-Wallis statistics reported in the Results section that I could find.
7. You emphasize the names of the locations rather than the names of the States. For someone not acquainted with the abbreviations, this is rather confusing. I wonder if it would be better to give the names of the states in full, just for clarity.
8. Suggest you give the names of the States in full when referring to a particular State.
9. You exposed the larvae to different concentrations, not doses. Indeed, the WHO now uses the term "discriminating concentrations", not diagnostic doses, although in the original 1991 document they did use the term "diagnostic dosages (concentrations)". This should be corrected to Diagnostic Concentrations (DC) throughout the manuscript. See: https://www.who.int/publications/i/item/9789240045200 for an up-to-date example albeit using adulticides.
10. Given that you only used 25 larvae x 4 replicates (100 insects per bioassay), the difference between a 97%, 98% or 99% result is just one or two larval deaths, which is a cause for caution that should be mentioned in the Discussion.
11. This makes it sound like the study [19] was the only study performed on commercial alternatives to temephos in Mexico. There are several other examples using insect growth regulators, spinosad, Bt, lambda-cyhalothrin and other compounds. I think you should broaden your reporting of examples here.
12. Temephos has proved to be efficient, but its great popularity is probably related to its very low COST. You can treat a lot of larval habitats with temephos granules for just a few dollars!
13. Suggest you reword this to "...continues to be one of the most widely used larvicides in tropical regions across the world".
14. Again, mg/L is a concentration, not a dose. (see point 9)
15. You do not really mention the main alternatives to temephos. What are the main alternative products being used by public health authorities in Mexico now?
16. The references should be formatted for Insects.

Well written in general. Minor editing, see scanned manuscript.
Reviewer 2 Report
Comments and Suggestions for Authors
Peer review report on the manuscript "Widespread resistance to temephos in Aedes aegypti (Diptera: Culicidae) from Mexico", (Manuscript ID: insects-2854318)
Recommendation: Accept
Comments to Authors:
This manuscript reports a study that evaluated 23 Aedes aegypti populations from Mexico for their susceptibility or resistance to temephos. The findings elucidate that, according to the criteria established by the World Health Organization (WHO), 22 out of the 23 populations exhibited a spectrum of resistance ranging from low to high intensity.
The manuscript is well-written with a well-organized text. The results are sufficiently presented and analyzed, and the discussion contributes to a better understanding of the findings. The study updates the knowledge of the resistance status of temephos and holds particular significance for the nation of Mexico, underscoring the imminent necessity for the implementation of strategic resistance management measures.
Therefore, the manuscript is recommended for publication in the Insects, considering the following minor comments.
The discussion could be improved by incorporating details regarding mosquito control programs in Mexico. Information such as the insecticides and especially the larvicides used, the frequency, and the type of spraying operations (aerial or ground) would be valuable to illuminate the potential factors contributing to variations in resistance rates across different regions.
In the lines 73, 100, 104 and 349 the scientific names of the organisms should be written in italics.
Reviewer 3 Report
Comments and Suggestions for Authors
Dear Authors, thank you for submitting an interesting article entitled „Widespread resistance to temephos in Aedes aegypti (Diptera: Culicidae) from Mexico” because the transfer of information is always useful.
Below you can find some general and specific comments.
This study aimed to analyze the frequency of temephos resistance in Aedes aegypti populations, a crucial element in mosquito control programs combating vector-borne diseases in Mexico. The authors conducted resistance assessments in twenty-three populations using varying doses of temephos, and the results indicate a concerning prevalence of resistance, especially at higher doses. The study suggests the necessity for a strategic rotation of larvicides to maintain effective control over the vector populations in the country.
General Comments:
The article provides valuable insights into the resistance patterns of Aedes aegypti populations to temephos. The focus on larvicides, particularly temephos, is relevant to the ongoing efforts in vector control.
Specific Comments:
The manuscript is generally clear and relevant to the field. However, the authors should consider expanding on the broader implications of high resistance levels, specifically in relation to the potential failure of control programs.
The experimental design is appropriate for the study's objectives.
The conclusions drawn are generally consistent with the evidence presented. However, a more explicit set of recommendations, especially regarding the proposed rotation of larvicides, would be valuable for practitioners and policymakers.
The manuscript is mostly clear and relevant to the field.
The references are relevant, and there is no apparent excessive self-citation.
The manuscript is scientifically sound.
In summary, the study provides valuable information on temephos resistance in Aedes aegypti populations in Mexico.
Round 2
Reviewer 1 Report
Comments and Suggestions for Authors
The authors have addressed my concerns and improved their manuscript.
Comments on the Quality of English LanguageMinor editing in journal production.